# CHANNEL-AWARE MIXED-PRECISION QUANTIZATION FOR EFFICIENT LONG-CONTEXT INFERENCE

**Chengxi Liao**[1]    **Zeyi Wen**[1,2*]
[1]The Hong Kong University of Science and Technology (Guangzhou)
[2]The Hong Kong University of Science and Technology
`cliao118@connect.hkust-gz.edu.cn, wenzeyi@hkust-gz.edu.cn`

## ABSTRACT

The key-value (KV) cache plays a vital role in accelerating autoregressive inference for large language models (LLMs). However, its linear memory growth with sequence length poses significant memory bottlenecks, especially in long-context scenarios. Quantization offers a promising solution for memory efficiency. While existing methods typically apply channel-wise quantization to the key cache and token-wise quantization to the value cache, they suffer from severe performance degradation under low-bit configurations. Our analysis reveals that quantization sensitivity varies across individual KV channels, presenting an opportunity for non-uniform bit allocation. Following this finding, we propose ChanMix, a mixed-precision quantization framework that supports channel-wise quantization on 2-bit setting with custom Triton kernels implementation. To improve low-bit quantization performance, we introduce a channel-aware bit reallocation strategy, which allocates bits across channel sensitivity. Through extensive evaluation, ChanMix demonstrates superior performance across the NIAH, RULER, and InfiniteBench benchmarks for the Llama, Mistral, and Qwen model families, achieving improvements of at least 5 absolute percentage points on RULER compared to all baseline methods. Additionally, ChanMix enables a 2.3× increase in batch size and supports a 1.5× longer context length during inference. Our code is available at `https://github.com/cxiliao/ChanMix`.

## 1 INTRODUCTION

Large language models (LLMs) have advanced rapidly in recent years, with growing demand for complex and longer input tasks (Shaham et al., 2023; Bai et al., 2025; Zhang et al., 2024). This has led to significant expansions in context length, making memory a critical bottleneck during inference. For instance, Llama-3.1 (Grattafiori et al., 2024) requires 125GB of memory per million tokens (Liu et al., 2024b), exceeding the memory capacity of most GPUs. Moreover, the memory footprint of KV cache restricts batch sizes, particularly for long-context scenarios. Efficient and effective KV cache compression thus remains a key challenge in LLM serving (Jiang et al., 2025).

Approaches to reducing KV memory overhead can be divided into three categories. The first category is using shared attention heads across the KV cache by modifying the attention mechanism (Shazeer, 2019; Ainslie et al., 2023; Liu et al., 2024a). While effective, these methods require architectural changes and pre-training or fine-tuning, limiting their universality. The second category is token eviction methods (Zhang et al., 2023; Li et al., 2024; Ge et al., 2024), which discard less important tokens based on scoring heuristics. Though memory-efficient, such methods incur significant performance degradation with permanent information loss. The third category is quantization methods (Yao et al., 2022; Liu et al., 2024d; Hooper et al., 2024; Yang et al., 2024b; Kang et al., 2024; Su et al., 2025b), which compress the KV cache by using uniform bit allocation across channels. While these quantization introduce minimal quality loss in general tasks, they still struggle with specialized scenarios like long-context retrieval, especially in low-bit settings.

ChanMix aims to boost the performance of quantization-based methods. We first investigate the distribution of KV cache channels. While prior works (Hooper et al., 2024; Liu et al., 2024d; Zhao

---

*Corresponding author.

et al., 2024) exploit outlier channels in the key cache to alleviate quantization error, we uncover that some robust channels with small-magnitude exist. This asymmetric sensitivity motivates us to allocate quantization bits in a channel-aware manner to reduce overall quantization error. We then locate one potential source of performance degradation in long-context scenarios. Recent studies (Wu et al., 2025; Tang et al., 2025; Xiao et al., 2025) have shown that certain channels are crucial for long-context retrieval. However, the impact of quantization on such channels remains unexplored. Our further experiments reveal that these retrieval-sensitive channels are highly vulnerable to quantization, and their bit precision disproportionately affects long-context performance. This finding explains the degradation of existing uniform quantization methods. Combining these two insights, we propose ChanMix , a channel-aware mixed-precision quantization framework for the KV cache. We further introduce a novel bit allocation strategy that adapts to channel sensitivity: for sensitive channels (i.e., retrieval and outlier channels), higher bit precision is assigned to mitigate performance degradation; for robust channels (i.e., small-magnitude channels), lower bit precision is used to save memory. This adaptive allocation enables ChanMix to significantly improve model performance while maintaining a low memory footprint. ChanMix is fully compatible with existing token pruning methods (e.g., (Xiao et al., 2025)) as well as the quantization error compensation methods (e.g., (Kang et al., 2024; Yankun et al., 2025)), making it a versatile solution for long-context KV cache compression. To summarize, our contributions include the following three key advancements:

- **ChanMix Framework:** A channel-wise mixed-precision quantization framework for KV cache that supports 2-bit, with channel reordering for efficient 8-bit-aligned packing and custom Triton kernel implementation.
- **Sensitivity-Aware Allocation:** A novel bit allocation strategy that leverages a three-way asymmetry in channel sensitivity—retrieval and outlier channels require higher precision, while small-magnitude channels can be quantized aggressively.
- **Extensive Evaluation:** Comprehensive experiments on Llama, Mistral, and Qwen across NIAH, RULER, and InfiniteBench show that ChanMix consistently improves accuracy over uniform quantization. Efficiency experiments further highlight our advantage by enabling a 2.3× increase in batch size and supporting a 1.5× longer context length.

## 2 RELATED WORK

### 2.1 LONG-CONTEXT LLMS

Long-context ability in LLMs becomes increasingly important as the demand for processing larger inputs grows. To extend the context length, one effective solution is fine-tuning the model with longer sequences. RoPE (Su et al., 2024) and ALiBi (Press et al., 2022) are two widely used methods which utilize relative position encoding, enabling the model to learn the relationships between tokens regardless of their absolute positions. Another solution is to use extra memory to restore the activation. ActivationBeacon (Zhang et al., 2025) introduces a well-trained compression module that condenses activation tokens into beacon tokens, and then retrieves the relevant tokens from these tokens for generation, allowing the model to process longer sequences without fine-tuning. Existing models like Llama-3.1 (Grattafiori et al., 2024) can reach 128K context length with RoPE, while Gemini (Kavukcuoglu, 2025) can achieve even 1M context length. The focus of long-context LLMs is shifting from extending context length to reducing the memory footprint of the KV cache. ChanMix aligns with this trend.

### 2.2 EFFICIENT LLMS INFERENCE

To reduce the inference cost of LLMs, the techniques can be broadly categorized into two directions:

**Model Compression**    Model compression techniques aim to reduce the memory footprint of model weights and accelerate inference speed. Singular Value Decomposition (SVD) is a widely adopted low-rank approximation method that reduces matrix dimensions by approximating a given matrix using two smaller matrices. Recent studies, such as ASVD (Yuan et al., 2023), SoLA (Huang et al., 2025), and SVD-LLM (Wang et al., 2025), have enhanced SVD by incorporating weight importance metrics to achieve a better trade-off between model accuracy and memory efficiency. Quantization represents another major line of work. Some approaches, including LLM-QAT (Liu et al., 2024c) and BitDistiller (Du et al., 2024), employ Quantization-Aware Training (QAT), retraining

models with quantization constraints to minimize error. In contrast, methods such as GPTQ (Frantar & Alistarh, 2023), AWQ (Lin et al., 2024), and SqueezeLLM (Kim et al., 2024) utilize Post-Training Quantization (PTQ), adjusting quantization parameters based on activation distribution to reduce error. ChanMix is fully compatible with these weight compression techniques, as it specifically targets the reduction of memory consumed by the KV cache.

**KV Cache Compression**   KV cache compression methods focus on decreasing activation memory during inference. A prominent strategy is token eviction. Recent work—including H2O (Zhang et al., 2023), StreamingLLM (Xiao et al., 2024), FastGen (Ge et al., 2024), SnapKV (Li et al., 2024), PyramidKV (Cai et al., 2024), and DuoAttention (Xiao et al., 2025)—observes that tokens contribute unequally to performance, and thus propose strategies to identify and prune less important tokens. However, such eviction approaches risk permanent information loss, particularly in multi-turn dialogue settings. Quantization has also been widely applied to KV cache compression. Methods like ZeroQuant (Yao et al., 2022), KVQuant (Hooper et al., 2024), KIVI (Liu et al., 2024d), and WKVQuant (Yue et al., 2024) use uniform bit-width quantization across the KV cache, often employing channel-wise grouping for keys and token-wise for values, significantly reducing quantization error. ChanMix is fully compatible with these token pruning methods. For quantization, uniform bit allocation overlooks varying channel sensitivity, potentially leading to suboptimal performance. ChanMix addresses this limitation by adaptively allocating quantization bits according to channel sensitivity.

**Mixed-Precision Quantization**   Numerous studies have employed mixed-precision techniques to enhance the inference efficiency of LLMs. For model weight compression, LLM-MQ (Li et al., 2023) proposes a layer-wise sensitivity-based precision allocation strategy for quantizing weight matrices. SliM-LLM (Huang et al., 2024) introduces a group-wise mixed-precision PTQ framework that assigns bit-widths according to weight salience. CMPQ (Chen et al., 2024) adopts a channel-wise quantization approach, adjusting precision based on activation distributions. PMPD (Chen et al., 2025) progressively reduces weight precision during inference to balance efficiency and performance. For KV cache compression, MiKV (Yang et al., 2024b), ZipCache (He et al., 2024), and OTT (Su et al., 2025a) identify salient or outlier tokens and allocate higher bit-widths to them. KVTuner (Li et al., 2025) enables sensitivity-aware layer-wise mixed-precision quantization for the KV cache. QAQ (Cheng et al., 2025) adaptively allocates quantization bit-width based on outlier characteristics and attention sensitivity. However, existing methods do not specifically address channel-sensitivity quantization of KV cache. ChanMix fills this gap.

## 3   BACKGROUND

In this section, we provide a brief overview of the transformer-based inference process and the concept of KV cache quantization.

### 3.1   INFERENCE PROCESS

The transformer-based (Vaswani et al., 2017) inference procedure consists of two distinct stages: pre-filling and decoding. In the pre-filling stage, the model processes the entire input prompt in one forward pass, computing and caching the KV states for all the tokens. This cached representation enables efficient attention computation during decoding. In the decoding stage, the model generates output tokens autoregressively. At each step of the decoding, the attention over the cached KV states is computed, and then the next token conditioned on all the previously generated tokens is predicted. Finally, the KV cache with the new token's hidden states is generated.

Formally, let $X_P \in \mathbb{R}^{b \times l \times d_m}$ denote the prompt embedding, where $b$ represents the batch size, $l$ the prompt length, and $d_m$ the model dimension. The query, key, and value weight matrices $W_q, W_k, W_v \in \mathbb{R}^{d_m \times d_h}$ project these embeddings, with $d_h$ indicating the hidden dimension. In the pre-filling stage, the initialization of $Q$, $K$, and $V$ is computed as:

$$Q = X_P W_q, \quad K = X_P W_k, \quad V = X_P W_v \tag{1}$$

During decoding with KV caching, for each input embedding $X_D \in \mathbb{R}^{b \times 1 \times d_m}$, the key and value states are updated via concatenation:

$$K \leftarrow \text{Concat}(K, X_D W_k), \quad V \leftarrow \text{Concat}(V, X_D W_v) \tag{2}$$

The attention mechanism then computes:

$$\text{Attention}(Q, K, V) = \text{Softmax}\left(\frac{QK^T}{\sqrt{d_h}}\right) V \qquad (3)$$

## 3.2 KV Cache Quantization

**Quantization Process**  To reduce the memory footprint of the KV cache, quantization techniques are commonly applied. Early approaches such as group quantization (Yao et al., 2022) divide the KV cache into smaller groups and apply quantization within each group to mitigate quantization error. More recent studies (Liu et al., 2024d; Hooper et al., 2024) investigate the distribution of KV activations and adopt finer-grained schemes, including channel-wise quantization for the key cache and token-wise quantization for the value cache, which have demonstrated promising results. Formally, let $C$ denote the KV cache and $b$ the number of quantization bits. For each group, we compute the zero point $Z = C_{min}$ and scale factor $S = (C_{max} - C_{min})/(2^b - 1)$. The quantized cache $\tilde{C}$ is derived as:

$$\tilde{C} = \text{Round}\left(\frac{C - Z}{S}\right), \quad C \approx S\tilde{C} + Z \qquad (4)$$

Here, $\tilde{C}$ is stored as a $b$-bit integer, while $Z$ and $S$ use full precision.

**Quantization Error**  For a given group, the maximum quantization error $E_{max}$ is defined as (detailed derivations are provided in the A.2)

$$E_{max} = \frac{S}{2}, \qquad (5)$$

which originates from both the rounding operation and the limited dynamic range of low-bit integers. Consequently, reducing the group scale factor $S$ is essential for mitigating quantization error.

## 4 Methodology

In this section, we first present our channel detection methods, then analyze the channel sensitivity of KV cache, and finally introduce our proposed ChanMix framework of KV cache quantization.

### 4.1 Channel detection

**Outlier and Subnormal Key Channel**  We first sort the channel range from offline collected key cache, according to the quantization error analysis 5. Let $K \in \mathbb{R}^{N \times C}$ denote the key cache matrix, where $N$ is the number of sample tokens and $C$ is the number of channels. For each channel $c$, we compute its dynamic range $R_c$ as:

$$R_c = \max_{1 \le i \le N} K_{i,c} - \min_{1 \le i \le N} K_{i,c} \qquad (6)$$

Then we use K-means clustering to cluster the channel ranges $\{R_1, R_2, \ldots, R_C\}$ into three groups: outlier, normal, and subnormal. To ensure these three types of channel are clustering correctly, we set the three initial centroids $\{R_{min}, R_{medium}, R_{max}\}$ for K-means clustering:

$$\mathcal{C}_{\text{low}}, \mathcal{C}_{\text{medium}}, \mathcal{C}_{\text{high}} \leftarrow \text{K-means}(R_1, \ldots, R_C, k = 3) \qquad (7)$$

Channels in $\mathcal{C}_{\text{high}}$ and $\mathcal{C}_{\text{low}}$ are designated as outliers and subnormal respectively.

**Retrieval Channel**  In attention-based LLMs, retrieval heads play a pivotal role in the model's retrieval capability, which is a critical phenomenon revealed by recent studies (Wu et al., 2025; Tang et al., 2025; Xiao et al., 2025; Lee et al., 2025; Donhauser et al., 2025). We propose a simple yet efficient method, different from the previous works. Our approach stems from the hypothesis that a model's long-context retrieval ability relies on capturing information between identical tokens across inputs (copy-paste operation (Wu et al., 2025)). So we construct a retrieval scenario to identify the retrieval heads. We first generate a semantically irrelevant sentence with $n$ tokens, and then repeat it $t$ times to create a sentence-level dependencies prompt. Here, $n$ and $t$ can be set to around 100 and 30, respectively, according to our experience. Then we modify the attention mask (cf. Figure 1) by removing local and sink attention scores (Xiao et al., 2024), eliminating intra-sentence semantic dependencies. Finally, we input the constructed prompt to the model to compute each head's retrieval score using the retrieval mask.

Formally, for sequence length $l$ and attention score matrix $A \in \mathbb{R}^{l \times l}$, we define the retrieval mask matrix $M \in \{0, 1\}^{l \times l}$. The retrieval score $S$ for an attention head is computed as:

$$S = \sum_{i}^{l} \sum_{j}^{l} A_{ij} \circ M_{ij} \tag{8}$$

where $\circ$ denotes the Hadamard product. Heads with higher aggregate scores are identified as retrieval heads, and channels in retrieval heads are defined as retrieval channels. Crucially, our method requires only one-shot inference, significantly reducing computational overhead compared to existing approaches.

### 4.2 CHANNEL SENSITIVITY ANALYSIS

**Outlier and Subnormal Key Channel** We analyzed key cache channels using 1,000 samples from WikiText-2 and compiled the results. Figure 2 (left) shows how the key cache values are distributed—subnormal channels are in blue, and outlier channels are in purple. There are

Figure 1: Our proposed A-shaped retrieval mask removes noise score from attention sinks and local tokens to isolate retrieval dependencies.

clearly fewer outliers than subnormal channels. Then we evaluate the sensitivity of these channels in quantization bit width on PPL metric. As Figure 2 (middle) shows, outlier channels become much worse with fewer bits, while subnormal channels stay stable. This phenomenon suggests that under the same bit-width budget, outlier channels should be allocated with more bits.

**Retrieval Channel** We also investigated the sensitivity of retrieval head channels to quantization bit-width. As shown in Figure 2 (right), increasing the bit allocation for retrieval channels leads to a more substantial improvement in retrieval accuracy, indicating that these channels are more sensitive to quantization performance.

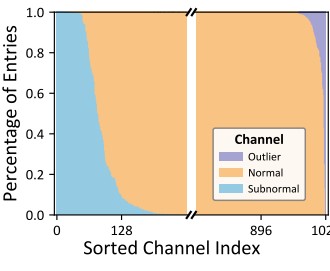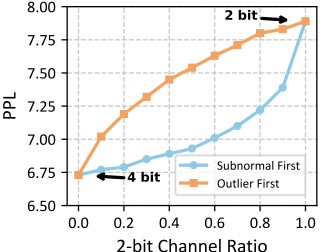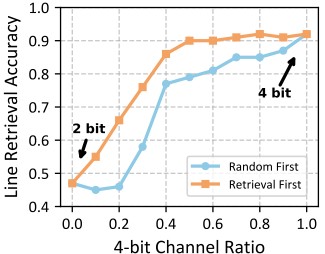

Figure 2: Left: Stacked distribution (%) of key cache values with Llama-3.1 on layer 20. Values are categorized as outlier (top 25% of max layer range), normal (middle 50%), and subnormal (bottom 25%). For instance, Channel 128 (right) exhibits 40% subnormal and 60% normal entries. Middle: The perplexity on WikiText-2, outlier channels are more sensitive to the decrease in the quantization bit. Right: The retrieval task accuracy, retrieval channels are more sensitive to the increase in the quantization bit on the retrieval task.

### 4.3 CHANMIX FRAMEWORK

**Overall Framework** ChanMix applies a straightforward quantization pipeline to the KV cache. During the prefill stage, after computing attention for each layer, the KV cache is quantized into low-bit format—comprising integer tensors, scale, and zero-point parameters—and stored in 8-bit aligned tensor. At the decoding stage, the quantized KV cache is dequantized back to full precision before attention computation.

**Sensitivity-Aware Bit Allocation** Following the analysis in Section 4.2, we propose two sensitive channel detection methods. As shown in Figure 3 (right), we allocate 4 bits to retrieval channels, 3

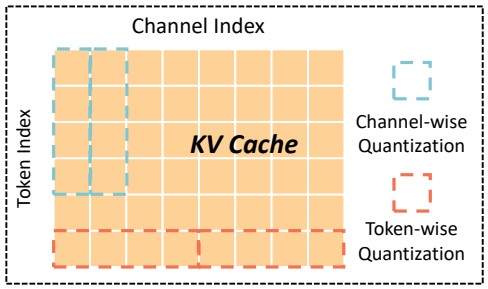 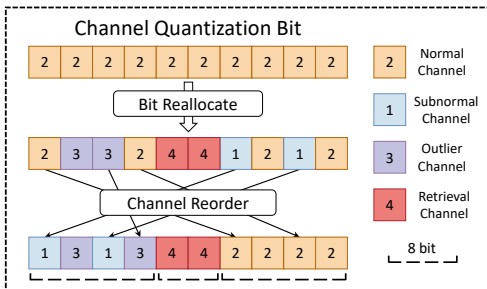

Figure 3: Left: channel-wise and token-wise quantization method. Right: bit reallocation for Chan-Mix (2-bit quantization). Reallocating channel bits by sensitivity: retrieval and outlier channels are allocated with more bits, while subnormal with less. Channel reordering ensures efficient 8-bit-aligned storage of the quantized cache.

bits to outlier channels, and 1 bit to subnormal channels. As shown in Section 4.2, the number of subnormal channels always exceeds that of outlier channels. Therefore, we reallocate only the same number of subnormal channels as outliers to maintain 8-bit–aligned storage.

**Implementation Details**   Following KIVI (Liu et al., 2024d), we apply channel-wise quantization to the key cache and token-wise quantization to the value cache, Figure 3 (left), and incorporate group-wise scaling and residual parameters to build our quantization framework. The quantized parameters are stored in the *float8_e4m3fnuz* format to minimize memory overhead. In the quantize-and-store step, a Triton kernel first efficiently quantizes the KV cache by calculating scale and zero-point parameters and converting the original cache into integer format. Another Triton kernel fuses reordering and storing operations to write the quantized KV cache into 8-bit aligned memory, minimizing memory copy overhead. During the read-and-dequantize step, a Triton kernel reads the quantized cache from the 8-bit aligned tensor and dequantizes it to full precision. This kernel also merges reordering and dequantization to reduce memory copy overhead. Note that our kernel implementation is fully compatible with FlashAttention, making it a more general solution.

## 5   EXPERIMENTS

In this section, We first present the experimental settings, then the results and analysis of long-context and short-context experiments, followed by an efficiency analysis and ablation studies.

### 5.1   SETTINGS

**Models**   We select Llama-2-7b-32k-Instruct (*Llama-2*) (together.ai, 2023) to represent the multi-head attention (MHA) model, and Llama-3.1-8b-Instruct (*Llama-3.1*, 128K context length) (Grattafiori et al., 2024), Llama-3-8B-Instruct-Gradient-1048k (*Llama-3*, 128k context length) (gradientai), Mistral-7b-Instruct-v0.3 (*Mistral*, 32K context length) (Jiang et al., 2023), and Qwen-2.5-14B-Instruct (*Qwen-2.5*, 32k context length) (Yang et al., 2024a) to represent the group-query attention (GQA) model, DeepSeek-R1-Llama-8B (*DS-R1-Llama*) (Guo et al., 2025) to represent reasoning model in our evaluation.

**Datasets**   We evaluate ChanMix on the following benchmark to assess the long-context capabilities. The Needle-In-A-Haystack (NIAH) benchmark (Kamradt, 2023) is a synthetic benchmark designed to evaluate a language model's capability to retrieve specific information from long-context scenarios. RULER benchmark (Hsieh et al., 2024) is a benchmark designed to evaluate long-context language models with an artificially generated synthetic dataset. InfiniteBench benchmark (Zhang et al., 2024) is designed to assess the ability of language models to handle long-context problems requiring processing, understanding, and reasoning over super long contexts (100K+ tokens) across real-world scenarios and synthetic scenarios. AIME is a mathematics reasoning benchmark featuring integer-answer problems that require multi-step symbolic derivations. While the input is short, it induces long reasoning outputs, evaluating sustained logical consistency and structured problem-solving ability. We also conduct experiments on the following short-context benchmark.

MMLU (Hendrycks et al., 2021) is a benchmark covering 57 subjects across STEM, humanities, and social sciences. MBPP (Austin et al., 2021) is a dataset of beginner-level Python programming tasks where models must generate correct functions that pass unit tests. GSM8K (Cobbe et al., 2021) is a math problem dataset focused on multi-step arithmetic reasoning.

**Configurations**    ChanMix is implemented using PyTorch and Triton for fused quantization kernel operations. To detect the outlier channels, we randomly sample 10 entries from the Wikitext-2-raw-v1 dataset (Merity et al., 2016) as the profiling dataset. Both outlier and retrieval channel offline profile processes can be done in 10 minutes on a single GPU. We uniformly apply the following settings across all the experiments: quantization bits for subnormal, normal, outlier, and retrieval channels are set to 1, 2, 3, and 4 bits, respectively, with a quantization group size of 32 and a residual length of 128 for both key and value caches. All the experiments are executed on a single NVIDIA H800 GPU with 80GB memory.

**Baseline methods**    To evaluate ChanMix's long-context capability, we compare it against five state-of-the-art KV cache compression methods: ZipCache (He et al., 2024), KIVI (Liu et al., 2024d), KVQuant (Hooper et al., 2024), DuoAttn (Xiao et al., 2025), and OTT (Su et al., 2025b). For fairness, all the methods (including the vanilla) use FlashAttention (Dao, 2024) as the unified attention backend. All baselines use their original paper's default configurations and set KV size to around 20%.

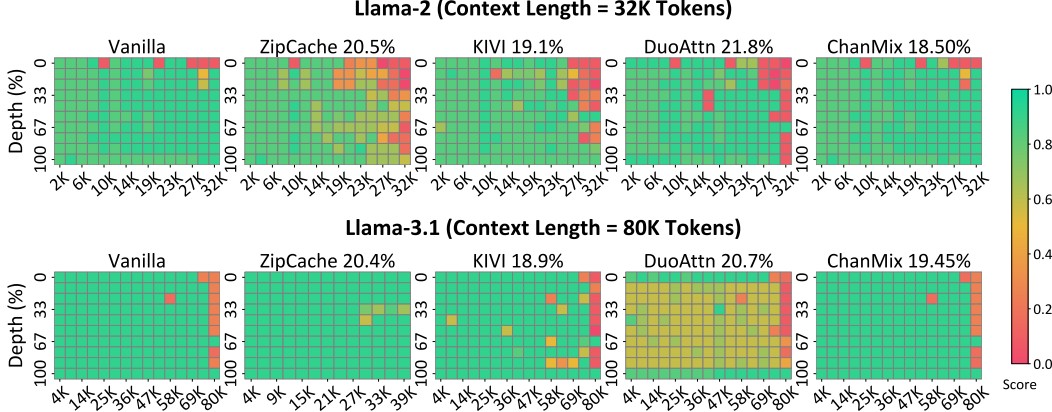

Figure 4: Needle-in-a-Haystack (NIAH) benchmark results for Llama-2 and Llama-3.1 models, compared with baseline approaches. ChanMix achieves the best for both model variants.

## 5.2 LONG-CONTEXT RESULTS

**NIAH**    As shown in Figure 4, ChanMix achieves near-vanilla model performance. Notably, it outperforms all the competing methods by a large margin, effectively bridging the accuracy gap between the vanilla model and other quantization methods. This aligns with our design goal of minimizing memory overhead with minimal accuracy loss.

**RULER**    As shown in Table 1, ChanMix maintains stable performance across diverse context lengths and models despite high compression ratios. Specifically, it incurs almost no accuracy degradation on Llama-2, Mistral, and Qwen models, reducing the memory consumption by 81%. In contrast, token-importance-based methods (e.g., ZipCache, OTT) exhibit significant accuracy drops, particularly for long contexts (longer than 16K). Quantization-based KIVI and KVQuant show milder degradation but fail to match ChanMix 's efficiency-accuracy tradeoff. These results conclusively demonstrate that ChanMix maximizes long-context performance while minimizing memory overhead.

**InfiniteBench**    Table 2 demonstrates that ChanMix achieves performance comparable to all the baselines across eight diverse tasks while significantly reducing KV cache size. These results validate that ChanMix maintains strong task-general capabilities while optimising memory efficiency.

Table 1: Comparison of ChanMix and other quantization methods on the RULER benchmark. The best results are highlighted in **bold**.

| Model | Method | KV Size | 4K | 8K | 16K | 32k | 64k | 128k | Average |
|---|---|---|---|---|---|---|---|---|---|
| Llama-2 32K | Vanilla | 100% | 80.36 | 75.89 | 61.42 | 53.82 | - | - | 67.87 |
| | ZipCache | 20.54% | 65.11 | 54.87 | 33.71 | 13.91 | - | - | 41.90 |
| | KIVI | 19.13% | 68.34 | 65.03 | 53.31 | 41.31 | - | - | 57.00 |
| | KVQuant | 19.77% | 70.84 | 66.28 | 59.43 | 49.6 | - | - | 61.54 |
| | DuoAttn | 21.81% | 59.07 | 51.38 | 41.69 | 23.38 | - | - | 43.88 |
| | ChanMix | **18.34%** | **80.35** | **74.03** | **63.82** | **53.20** | - | - | **67.85** |
| Mistral 32K | Vanilla | 100% | 91.66 | 87.64 | 86.96 | 81.69 | - | - | 86.99 |
| | ZipCache | 20.62% | 83.31 | 74.35 | 64.57 | 30.66 | - | - | 63.22 |
| | KIVI | **19.19%** | 88.52 | 83.20 | 78.07 | 67.01 | - | - | 79.20 |
| | OTT | 19.40% | 77.49 | 70.14 | 65.57 | 55.17 | - | - | 67.09 |
| | ChanMix | 19.59% | **91.55** | **87.51** | **86.37** | **79.98** | - | - | **86.35** |
| Llama-3 128K | Vanilla | 100% | 88.73 | 87.89 | 81.86 | 76.86 | 71.59 | 70.2 | 79.52 |
| | KVQuant | 19.77% | 83.88 | 81.71 | 73.08 | 65.79 | 62.05 | 62.18 | 71.45 |
| | ChanMix | **19.45%** | **87.8** | **86.4** | **81.75** | **76.15** | **71.7** | **70.01** | **78.97** |
| Llama-3.1 128K | Vanilla | 100% | 93.90 | 85.18 | 86.21 | 84.66 | 82.26 | 74.59 | 84.47 |
| | ZipCache | 20.39% | 87.69 | 76.31 | 68.49 | 60.67 | OOM | OOM | 73.29 |
| | KIVI | **18.90%** | 85.42 | 80.10 | 76.07 | 72.58 | 68.36 | OOM | 76.51 |
| | DuoAttn | 20.73% | 73.11 | 67.36 | 62.33 | 60.82 | 55.99 | 34.58 | 59.03 |
| | OTT | 18.98% | **90.77** | 84.19 | 52.44 | 2.73 | 0.00 | OOM | 46.03 |
| | ChanMix | 19.45% | 90.64 | **86.01** | **83.66** | **82.68** | **80.55** | **69.41** | **82.16** |
| Qwen-2.5 32K | Vanilla | 100% | 97.27 | 95.53 | 94.11 | 92.17 | - | - | 94.77 |
| | ChanMix | 19.45% | 97.3 | 95.68 | 94.14 | 91.4 | - | - | 94.63 |

Table 2: Comparison of ChanMix and other quantization methods on the InfiniteBench benchmark. The **bold**/underlined numbers indicate the first/second highest value in each column.

| Model | Method | KV Size | En.Sum | En.QA | En.MC | En.Dia | Zh.QA | Code.Debug | Math.Find | PassKey | Average |
|---|---|---|---|---|---|---|---|---|---|---|---|
| Llama-2 32K | Vanilla | 100% | 20.50 | 12.70 | 40.17 | 5.00 | 18.44 | 25.63 | 13.14 | 27.12 | 20.34 |
| | KIVI | 18.95% | 17.41 | 13.72 | **43.67** | **7.50** | 18.77 | 25.89 | 11.71 | 21.19 | **19.98** |
| | KVQuant | 19.76% | 17.57 | 11.51 | 41.05 | 6.50 | 16.19 | **27.66** | 12.29 | 26.95 | 19.97 |
| | DuoAttn | 20.96% | 13.50 | **16.91** | 39.74 | 0.00 | 11.69 | 20.05 | 0.86 | 15.76 | 14.81 |
| | ChanMix | 18.23% | **17.77** | 13.73 | 41.05 | 6.50 | **19.09** | 25.63 | 12.57 | 22.37 | 19.84 |
| Mistral 32K | Vanilla | 100% | 23.31 | 9.39 | 51.09 | 8.50 | 22.99 | 28.68 | 25.14 | 27.12 | 24.53 |
| | KIVI | 18.95% | **22.82** | **9.06** | 50.66 | 7.00 | **23.55** | 28.43 | **26.57** | 27.12 | **24.40** |
| | OTT | 19.07% | 21.84 | 8.91 | 51.09 | 7.00 | 21.72 | 28.68 | 26.00 | 27.12 | 24.05 |
| | ChanMix | 19.48% | 21.93 | 8.66 | 51.09 | 7.50 | 23.43 | 29.19 | 26.00 | 27.12 | 24.37 |
| Llama-3 128K | Vanilla | 100% | 21.24 | 10.40 | 68.12 | 5.00 | 11.18 | 24.87 | 36.86 | 100.00 | 34.71 |
| | KVQuant | 19.76% | **23.75** | **10.00** | 66.81 | 4.50 | 11.37 | 24.87 | 36.86 | 100.00 | 34.77 |
| | ChanMix | 19.41% | 23.02 | 9.97 | 66.81 | 5.00 | 12.22 | 24.87 | 36.57 | 100.00 | **34.81** |
| Llama-3.1 128K | Vanilla | 100% | 26.97 | 10.31 | 65.50 | 20.50 | 13.45 | 23.35 | 33.43 | 100.00 | 36.69 |
| | DuoAttn | 20.26% | 19.67 | 8.12 | **67.69** | 11.00 | 12.49 | 22.84 | **34.00** | 84.75 | 32.57 |
| | ChanMix | 19.41% | **24.23** | 9.41 | 65.94 | 16.50 | 12.93 | 23.35 | 33.43 | 100.00 | **35.72** |
| Qwen-2.5 32K | Vanilla | 100% | 24.32 | 6.73 | 59.39 | 19.00 | 8.44 | 37.56 | 34.29 | 27.12 | 27.11 |
| | ChanMix | 19.41% | 24.66 | 7.01 | 60.70 | 21.50 | 8.64 | 37.56 | 33.43 | 27.12 | 27.58 |

**AIME**    Table 3 presents the performance comparison on the AIME24&25 benchmarks. ChanMix maintains comparable performance to other baseline methods. Notably, all three quantization-based methods show performance degradation on these long-generation tasks, indicating that low bit KV cache quantization remains a key challenge for sustained reasoning.

Table 3: Comparison of ChanMix and other methods on AIME benchmark with pass@1 accuracy.

| Method | AIME24 | AIME25 |
|--------|--------|--------|
| Vanilla | 43.33 | 23.33 |
| KIVI | 26.67 | 16.67 |
| OTT | 16.67 | 20 |
| ChanMix | 26.67 | 20 |

Table 4: Comparison of ChanMix and KIVI on the MMLU, MBPP, GSM8K benchmark.

| Model | Method | MMLU | MBPP | GSM8K |
|-------|--------|------|------|-------|
| Mistral | Vanilla | 60.17 | 36.4 | 43.59 |
| Mistral | KIVI | 59.82 | 29.4 | 39.19 |
| Mistral | ChanMix | 60.15 | 33.4 | 42.46 |
| Llama-3.1 | Vanilla | 62.77 | 47.4 | 76.19 |
| Llama-3.1 | KIVI | 62.57 | 44.2 | 74.75 |
| Llama-3.1 | ChanMix | 62.54 | 47.2 | 73.54 |

## 5.3 Short-Context Results

We compare ChanMix with KIVI on three downstream short-context tasks: MMLU, MBPP, and GSM8K, covering general knowledge, code, and mathematical reasoning domains. As shown in Table 4, ChanMix performs better than KIVI in most settings. These results demonstrate the strong generalizability of ChanMix across diverse downstream domains and short-context scenarios.

## 5.4 Efficiency Analysis

We evaluate ChanMix's efficiency by comparing it with a strong baseline, KIVI, and the vanilla model in three experiments. The memory presented is peak memory usage collected during inference with PyTorch tools. In Figure 5 (left and middle), we construct a serving scenario (128 prefilling tokens + 368 decoding tokens), then increase the batch size. With our fused quantization and dequantization kernels, ChanMix outperforms KIVI with larger throughput and lower memory usage, enabling 2.3× batch size than the vanilla model under the same memory. In Figure 5 (right), we fix the batch size to 1 and progressively increase the context length until out of memory (OOM) occurs. ChanMix supports 1.5× longer contexts than the vanilla model while consuming less memory. In contrast, the implementation of KIVI appears to contain issues that might affect its long prefilling performance. These experiments show that our fused kernels significantly reduce the memory copy overhead, achieving a state-of-the-art performance in KV quantization.

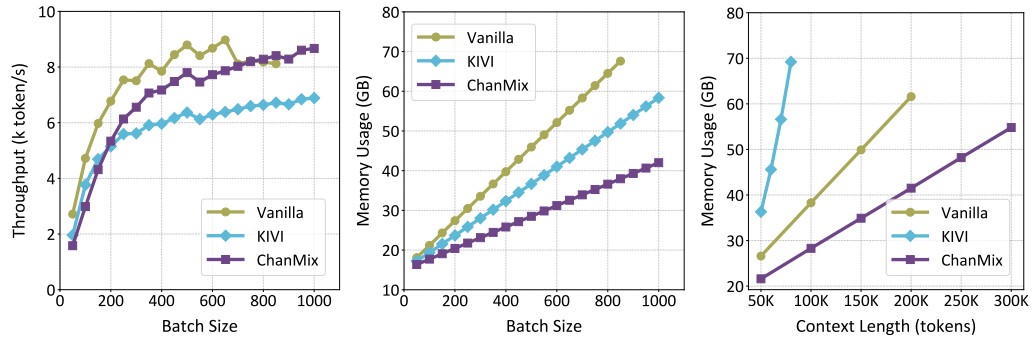

Figure 5: Throughput and memory usage. ChanMix can enable a larger context length and batch size with lower memory usage and achieve the best performance compared with KIVI.

## 5.5 Ablation Study

**Bit allocation strategy** We conduct ablation studies to evaluate the effectiveness of two sensitivity-aware bit allocation strategies. We use the RULER benchmark with LLaMA-3.1 and Mistral (Table 5) as the experiment setting. We begin with the base variant (ChanMix2), which applies 2-bit KV cache quantization without any enhancement: pure 2-bit KV cache quantization suffers significant accuracy degradation. Adding the retrieval channel component (ChanMix2+R) improves accuracy substantially while maintaining compression efficiency. The independent outlier

Table 5: Ablation study of ChanMix by combining two types of sensitivity channels components.

| Method | KV Size | Mistral | KV Size | Llama-3.1 |
|---|---|---|---|---|
| Vanilla | 100% | 86.99 | 100% | 84.47 |
| ChanMix2 | 16.06% | 72.13 | 15.78% | 75.40 |
| ChanMix2+R | 19.59% | 86.12 | 19.45% | 81.50 |
| ChanMix2+O | 16.06% | 84.17 | 15.78% | 81.13 |
| ChanMix2+R+O | 19.59% | **86.35** | 19.45% | **82.16** |

Table 6: Ablation study of ChanMix with different G and R on RULER with Lama-3.1 model.

| G | R | ChanMix2 | ChanMix | G | R | ChanMix2 | ChanMix | G | R | ChanMix2 | ChanMix |
|---|---|---|---|---|---|---|---|---|---|---|---|
| 32 | 32 | 81.88 | 83.92 | 64 | 32 | 76.58 | 82.22 | 128 | 32 | 69.78 | 78.02 |
| 32 | 64 | 82.24 | 84 | 64 | 64 | 76.68 | 82.69 | 128 | 64 | 70.13 | 78.91 |
| 32 | 128 | 82.2 | 83.87 | 64 | 128 | 76.75 | 82.67 | 128 | 128 | 70.01 | 78.96 |

compensation variant (ChanMix2+O) also boosts accuracy without memory overhead. These results indicate that both enhancements are effective and independent. When combining both components (ChanMix2+R+O), we observe the best overall performance: 82.16 on Llama-3.1 and 86.35 on Mistral, which is remarkably close to the full-precision results, confirming the complementary nature of the two enhancements and the effectiveness of the full ChanMix framework.

**Group size and residual length** We further conduct experiments with different combinations of group size $G$ and residual length $R$ on the RULER benchmark. Table 6 shows that across nine configurations, ChanMix consistently outperforms ChanMix2, indicating that the observed performance gains stem from our proposed bit allocation strategy rather than specific choices of group size or residual length. Moreover, ChanMix effectively mitigates the accuracy degradation under uniform low-bit quantization settings.

## 6 CONCLUSION

We have presented ChanMix , a channel-aware mixed-precision quantization framework for efficient long-context KV cache compression. Our framework starts from a novel aspect that allocates quantization bits by channel sensitivity, resulting in effective performance utilization of each quantization bit. With our bit allocation strategy, ChanMix achieves near-full-precision performance on the RULER benchmark, yielding at least a 5 percentage-point absolute improvement over all baselines and models, while using similar memory. Efficiency results show that our custom Triton kernels support practical serving settings, allowing 2.3× larger batch sizes and 1.5× longer context lengths compared to the vanilla model. ChanMix consistently surpasses KIVI in both accuracy and efficiency, establishing a stronger KV quantization baseline.

## ACKNOWLEDGMENTS

This work is supported by National Key R&D Program of China under Grant No.2024YFA1012700. It is also funded by the NSFC Project (No. 62306256) and the Natural Science Foundation of Guangdong Province (No. 2025A1515010261).

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

## A APPENDIX

### A.1 LLM USAGE DECLARATION

We use LLMs for grammar correction and text refinement.

### A.2 THEORETICAL DERIVATIONS FOR QUANTIZATION ERROR

For a channel group, asymmetric quantization is performed by computing the zero point $Z = C_{\min}$ and scale $S = (C_{\max} - C_{\min})/(2^b - 1)$. The quantized cache is

$$\tilde{C} = \text{Round}\left(\frac{C - Z}{S}\right), \ C \approx S\tilde{C} + Z \tag{9}$$

Let $x = (C - Z)/S$, the quantization error $E$ is defined as:

$$
\begin{aligned}
E &= C - (S\tilde{C} + Z), \\
&= C - \left(S \cdot \text{Round}\left(\frac{C-Z}{S}\right) + Z\right), \\
&= S\left(x - \text{Round}(x)\right), \\
&\leq \frac{S}{2}
\end{aligned}
\tag{10}
$$

Therefore, the maximum quantization error $E_{\max}$ is:

$$E_{\max} = \frac{S}{2} \tag{11}$$

### A.3 EXPERIMENTAL DETAILS

### A.3.1 EXPERIMENT SELECTION

Due to practical limitations, certain baselines were excluded or partially evaluated, as outlined below:

- **ZipCache:** ZipCache encounters OOM errors for contexts exceeding 38k tokens. Consequently, we evaluated it only on benchmarks with contexts under 32k tokens. We further excluded InfiniteBench due to its prohibitive execution time (estimated to be greater than one month per GPU).
- **KIVI:** KIVI also fails with OOM beyond 64k tokens. Thus, we restricted its evaluation to benchmarks with contexts below this threshold.
- **DuoAttn:** DuoAttn currently lacks support for the Mistral architecture. So, we only evaluated it on Llama-2 and Llama-3.1 models.
- **KVQuant:** KVQuant currently only support Llama-2 and Llama-3 models, so we only evaluate its performance on these two models.

### A.3.2 KV SIZE CALCULATION

Table 7 summarizes the average token lengths across different benchmarks for the evaluated models:

Table 7: Average token lengths across benchmarks.

| Model | NIAH | RULER | InfiniteBench |
|---|---|---|---|
| Llama-2 | 17,000 | 14,874 | 32,000 |
| Mistral | 17,000 | 14,675 | 32,000 |
| Llama-3.1 | 42,000 | 42,341 | 116,429 |

Given batch size $b$, hidden dimension $h$, token length $l$, residual length $r$, and group size $g$, we derive the KV cache size under 2-bit quantization for Llama-3 with several baseline methods as follows:

- **ZipCache:** With 60% salient token ratio and $r = 100$:

$$R_{\text{ZipCache}} = \frac{2blh \cdot 2 \cdot 1.6 + 3h \cdot 16 + 2bl \cdot 16 + 2b \cdot \frac{r}{2} \cdot h \cdot 16}{2blh \cdot 16}$$

- **KIVI:** With $g = 32$ and $r = 128$:

$$R_{\text{KIVI}} = \frac{2blh \cdot 2 + \frac{2 \cdot 2blh \cdot 16}{g} + 2b \cdot \frac{r}{2} \cdot h \cdot 16}{2blh \cdot 16}$$

- **DuoAttn:** With 20% sparsity, a sink size of 128, and 256 recent tokens:

$$R_{\text{DuoAttn}} = \frac{2b(128 + 256)h \cdot 16 \cdot 0.8 + 2blh \cdot 16 \cdot 0.2}{2blh \cdot 16}$$

- **ChanMix:** With 30% retrieval head enhancement, $g = 32$, and $r = 128$:

$$R_{\text{ChanMix}} = \frac{2blh \cdot 2 \cdot 1.3 + \frac{2 \cdot 2blh \cdot 8}{g} + 2b \cdot \frac{r}{2} \cdot h \cdot 16}{2blh \cdot 16}$$

Here, each $R$ represents the normalized memory ratio compared to a full-precision KV cache, under the specific configurations of each method.

## A.4 KEY VALUE CACHE VISUALIZATION

Figure 6 shows that in key cache, some fixed channels exhibit very large magnitudes, whereas in value cache, there is no significant pattern for outliers, this phenomenon suggests that we can apply channel-wise quantization for key cache, while it's not necessary for value cache.

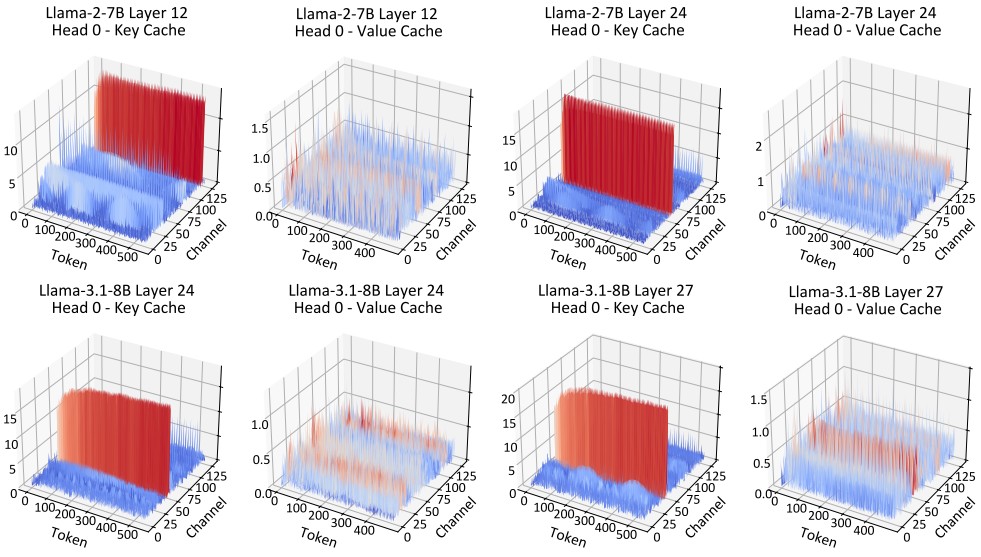

Figure 6: Magnitude of key and value cache for Llama-2-7B-32K and Llama-3.1-8B-Instruct.

Table 8: Results on large size models.

| Model | Method | KV Size | niah_s1 | niah_s2 | niah_s3 | niah_m1 | niah_m2 | niah_m3 | niah_mv | niah_mq | vt | fwe | cwe | qa_1 | qa_2 | Average |
|---|---|---|---|---|---|---|---|---|---|---|---|---|---|---|---|---|
| Qwen-2.5 32K | Vanilla | 100% | 100 | 100 | 100 | 98 | 92 | 98.5 | 99.75 | 98.12 | 99.8 | 88.67 | 95.7 | 71 | 66.5 | 92.92 |
| Qwen-2.5 32K | ChanMix | 17.91% | 100 | 100 | 100 | 98 | 91.5 | 98 | 99.25 | 98.12 | 98.2 | 87.33 | 94 | 68.5 | 62.5 | 91.95 |
| Llama-3 8K | Vanilla | 100% | 100 | 100 | 100 | 100 | 100 | 98 | 99.5 | 99.5 | 100 | 96.7 | 99.4 | 82 | 76 | 96.2 |
| Llama-3 8K | ChanMix | 19.78% | 100 | 100 | 100 | 100 | 100 | 98 | 99 | 99.5 | 100 | 94 | 98.8 | 84 | 76 | 96.1 |

## A.5 RESULTS ON LARGE MODELS

We conduct experiments on RULER benchmark with two large size models: Qwen-2.5-32B and Llama-3.1-70B. Results on table 8 show that ChanMix can perform well on these large size models.

