# OpenReview forum: "Channel-Aware Mixed-Precision Quantization for Efficient Long-Context Inference"
_ICLR.cc/2026/Conference — ICLR 2026 Poster_

### Official Review · Reviewer_HDr3 · 2025-10-28

**Soundness:** 3
**Presentation:** 3
**Contribution:** 3
**Rating:** 4
**Confidence:** 4

**Summary:**

This paper studies KV cache compression, which is important for LLM serving in real-world applications, especially in long-context scenarios. The authors claim that uniform bit allocation is the source of error for methods applying channel-wise quantization of keys and token-wise quantization of values. The authors propose ChanMix, a mixed-precision outlier-aware framework specifically designed for KV cache quantization. They evaluate their method on several benchmarks and develop a custom Triton kernel implementation to support their claims.

**Strengths:**

1.	The paper is well written. The tables and figures are well presented.
2.	The numerical results (e.g. Table 1) are promising: ChanMix outperform several existing methods in most cases.
3.	The efficiency analysis is clear, and a kernel implementation is useful for the community.

**Weaknesses:**

1. Lines 17–19 and 53 overstate the findings. While the data is compelling, it does not support the claim that all quantization error stems from uniform bit-width allocation. I recommend softening the language to allow for nuance. Mixed-precision quantization is not new, and sensitivity variation across channels is expected. That said, this appears to be the first application of mixed-precision to KV, which is appreciated. The paper is well-motivated, but the novelty and breadth of the claims should be tempered.
2. While ‘outlier’ is a widely used term in quantization literature, retrieval channel is a more recent, niche idea emerging from interpretability studies of attention heads, not a canonical term in the context of quantization. The corresponding part in section 4.1 should be expanded to provide more definition and background as retrieval channel is important in the motivation and implementation of the method proposed in this paper.

**Questions:**

1. How are outliers and subnormal channels defined? The first usage is Section 4.1 without proper definition.
2. Can you please provide more explanation for the statement ‘Channel reordering ensures efficient 8-bit aligned storage of the quantized cache’? It is also not clear to me how this reordering is done.
3. The largest models the paper has experiments with are Llama-3-8B and Qwen-2.5-14B. It would be good to test on larger models, such as Qwen3-32B, to see if the method scales well. Since sensitivity-aware bit allocation is a key contribution, I think the paper would benefit from adding QAQ into the comparison. Have the authors done such experiments?

I am open to increasing my rating if the weaknesses and questions are resolved.

---

> ### Author Response · Authors · 2025-11-25
> **Response to Reviewer HDr3**
>
> We sincerely thank the reviewer for the insightful feedback and constructive suggestions. Below, we address the concerns in more detail.
>
> > **Q1** The paper is well-motivated, but the novelty and breadth of the claims should be tempered (lines 17-19 and 53).
>
> Thank you for the helpful suggestion. We agree that the original wording overstated the findings. We have revised the sentences in lines 17–19 and 53 in the updated PDF to soften the claims and better reflect the nuance of our results.
>
> > **Q2** The corresponding part in section 4.1 should be expanded to provide more definition and background as retrieval channel is important in the motivation and implementation of the method proposed in this paper.
> >
> > **Q3** How are outliers and subnormal channels defined? The first usage is Section 4.1 without a proper definition.
>
> We have added additional background on the retrieval channel and reorganized the content in Section 4 of the updated PDF.
>
> To categorize channels, we perform K-means clustering on the key cache channel ranges and classify them into subnormal, normal, and outlier channels. For detailed definitions of subnormal and outlier channels, please refer to Section 4.1 in the updated PDF.
>
> > **Q4** Can you please provide more explanation for the statement 'Channel reordering ensures efficient 8-bit aligned storage of the quantized cache'?
>
> Please refer to Figure 3 (Right). The top row shows the original bit distribution under 2-bit quantization. After bit reallocation, some channels receive different bit-widths. If we store the quantized cache in this channel order, the bit sequence becomes irregular. For example, the first four channels (2,3,3,2 bits) sum to 10 bits, which requires two 8-bit words and wastes 6 bits storage.
>
> By introducing channel reordering (the bottom row), we rearrange channels so their bit-width patterns pack cleanly into 8-bit boundaries. This ensures that the entire quantized KV cache can be stored using 8-bit-aligned words, avoiding fragmentation and improving memory efficiency.
>
> > **Q5** It would be good to test on larger models, such as Qwen3-32B, to see if the method scales well.
>
> We evaluate ChanMix on the RULER benchmark using the Meta-Llama-3.1-70B-Instruct model with 8k context length and Qwen2.5-32B-Instruct model with 32k context length. ChanMix still performs well on these large size models.
>
> |               | Method  | KV Size | niah_s1 | niah_s2 | niah_s3 | niah_m1 | niah_m2 | niah_m3 | niah_mv | niah_mq |  vt   |  fwe  | cwe  | qa_1 | qa_2 | Avg   |
> | :-----------: | :-----: | :-----: | :-----: | :-----: | :-----: | :-----: | :-----: | :-----: | :-----: | :-----: | :---: | :---: | :--: | :--: | :--: | ----- |
> | Qwen2.5 (32K) |  FP16   |  100%   |   100   |   100   |   100   |   98    |   92    |  98.5   |  99.75  |  98.12  | 99.8  | 88.67 | 95.7 |  71  | 66.5 | 92.92 |
> | Qwen2.5 (32K) | ChanMix | 17.91%  |   100   |   100   |   100   |   98    |  91.5   |   98    |  99.25  |  98.12  | 98.2  | 87.33 |  94  | 68.5 | 62.5 | 91.95 |
> |  Llama3 (8K)  |  FP16   |  100%   |  100.0  |  100.0  |  100.0  |  100.0  |  100.0  |  98.0   |  99.5   |  99.5   | 100.0 | 96.7  | 99.4 | 82.0 | 76.0 | 96.2  |
> |  Llama3 (8K)  | ChanMix | 19.78%  |  100.0  |  100.0  |  100.0  |  100.0  |  100.0  |  98.0   |  99.0   |  99.5   | 100.0 | 94.0  | 98.8 | 84.0 | 76.0 | 96.1  |
>
> > **Q6** I think the paper would benefit from adding QAQ into the comparison.
>
> We reviewed the open-source implementation of QAQ. However, we found that the inference speed is slow since they didn't implement highly optimized custom kernels, and it's impossible for us to evaluate QAQ in our experiment settings.
>
> Based on our analysis of the QAQ methodology, we expect its performance to be similar to KVQuant, since both approaches preserve full-precision outliers in the KV cache and use dynamic bit allocation for the Key and Value caches. We have added the discussion with QAQ on the related work in the updated PDF file.

---

> > ### Comment · Reviewer_HDr3 · 2025-11-25
> >
> > Thank you for your detailed responses. Most of my concerns have been addressed, but I do have one remaining concern regarding the interpretation of your results. The current data still does not fully support the claim that all quantization error stems from uniform bit-width allocation. I suggest rephrasing these statements to align with what your experiments demonstrate. For example, in lines 17–19, you write:
> > >“Our analysis reveals that this degradation stems from uniform bit allocation across channels, as different KV channels contribute variously to model performance.”
> >
> > A more precise phrasing could be:
> > >“Our analysis reveals that quantization sensitivity varies across individual KV channels, presenting an opportunity for non-uniform bit allocation.”
> >
> > Ensuring claims are stated accurately is important, especially as this area is gaining attention. Clarity will help reduce noise in the literature.

---

> > > ### Author Response · Authors · 2025-11-26
> > > **Replying to Response to Reviewer HDr3**
> > >
> > > Thank you for your detailed and patient feedback. We sincerely apologize for the oversight that we updated our responses earlier, but forgot to upload the revised PDF to OpenReview. The new version has now been uploaded.
> > >
> > > Following your suggestion, we have revised the statements on lines 17–19 and line 54 so that they more accurately reflect our experimental findings. You can find the updated wording in the latest PDF. Thank you again for helping us improve the clarity and precision of our paper.

---

> > > > ### Comment · Reviewer_HDr3 · 2025-11-27
> > > >
> > > > Thank you for your responses and revision. I am now inclined to recommend acceptance of this paper. The contributions are a welcomed addition to the community.

---

> > > > > ### Author Response · Authors · 2025-11-27
> > > > > **Replying to Reviewer HDr3**
> > > > >
> > > > > Thank you for your positive evaluation and appreciation of our paper and responses. Your suggestions helped us refine several statements and significantly improved the clarity and readability of the paper. We sincerely appreciate the time and effort you have dedicated to our work.

---

### Official Review · Reviewer_oLua · 2025-10-30

**Soundness:** 2
**Presentation:** 2
**Contribution:** 2
**Rating:** 2
**Confidence:** 4

**Summary:**

The paper proposes ChanMix, a channel-aware mixed-precision quantization framework for compressing the key-value cache to enable efficient long-context inference. It identifies asymmetric sensitivities across KV channels, categorizing them as outlier, subnormal, and retrieval-sensitive, and allocates bits adaptively (1-4 bits) to minimize quantization error while maintaining low memory usage. The framework supports 2-bit to FP8 precision, includes channel reordering for 8-bit-aligned packing, and uses custom Triton kernels for implementation (but the code is not provided). Evaluations on Llama, Mistral, and Qwen models across NIAH, RULER, and InfiniteBench benchmarks show at least 5% improvement on RULER over baselines, with 2.3× larger batch sizes and 1.5× longer contexts.

**Strengths:**

- The paper's originality stems from its novel three-way channel sensitivity categorization (outlier, subnormal, retrieval) and adaptive bit allocation (4 bits for retrieval, 3 for outliers, 1 for subnormals; Figure 3, lines 290-294).
- This assessment is based on the methodology (Section 4, lines 162-323) and claims in the abstract/introduction, emphasizing how it removes limitations from prior quantization works (e.g., uniform bits in KIVI, line 119) while the setting is very limited.

**Weaknesses:**

- Code is not provided, resulting in a lack of reproducibility.
- The experimental setup does not clearly demonstrate the effectiveness of the proposed method. In `L-323`, a group size of 32 and a residual length of 128 are mentioned. The group size is relatively small, while the residual length is large. This configuration could be used to show the method’s effectiveness under less restrictive settings.
- The retrieval-head detection hyperparameters (n, t) are fixed “by experience.” However, stability across different values and architectures is not explored

**Questions:**

- Please address the items mentioned under Weaknesses. For example:
a. Lack of reproducibility
b. Use of limited settings for group size and residual length

- Your evaluation focuses mainly on long-context input-short-context output datasets. What is the performance of your method on long-context output tasks?
- There is ambiguity in the “≥5% improvement” claim. The abstract states “improvements of at least 5% on RULER,” but the tables show absolute percentage-point gains compared to the baselines (e.g., +5-8 points). Please clarify whether “%” refers to percentage points or relative percent.

---

> ### Author Response · Authors · 2025-11-25
> **Response to Reviewer oLua**
>
> We sincerely thank the reviewer for the insightful feedback and constructive suggestions. Below, we address the concerns in more detail.
>
> > **Q1** Code is not provided, resulting in a lack of reproducibility.
>
> We have provide the source code on this [link][https://anonymous.4open.science/r/ChanMix], please follow the Readme.md file to reproduce the experiment results.
>
> > **Q2** Use of limited settings for group size and residual length.
>
> We add more ablation experiments on Llama-3.1-8B-Instruct using the RULER-32K benchmark. ChanMix2 refers to standard 2-bit KV quantization, while ChanMix is our proposed channel-aware mixed-precision method. The results show that ChanMix effectively mitigates the accuracy drop in uniform low-bit settings quantization scenario on various group size and residual length.
>
> | Group Size | Residual Size | ChanMix2 Acc | ChanMix Acc |
> | :--------: | :-----------: | :----------: | :---------: |
> |     32     |      32       |    81.88     |  **83.92**  |
> |     32     |      64       |    82.24     |  **84.00**  |
> |     32     |      128      |     82.2     |  **83.87**  |
> |     64     |      32       |    76.58     |  **82.22**  |
> |     64     |      64       |    76.68     |  **82.69**  |
> |     64     |      128      |    76.75     |  **82.67**  |
> |    128     |      32       |    69.78     |  **78.02**  |
> |    128     |      64       |    70.13     |  **78.91**  |
> |    128     |      128      |    70.01     |  **78.96**  |
>
> > **Q2** What is the performance of your method on long-context output tasks?
>
> We evaluate ChanMix on the AIME benchmark using the DeepSeek-R1-Distill-Llama-8B model and report the Pass@1 accuracy. The results show that ChanMix consistently outperforms other baselines on long-context generation tasks.
>
> We believe the remaining accuracy gap can be addressed by incorporating additional sensitivity channel such as the reasoning channel [1], which is crucial for reasoning tasks. ChanMix is flexible and can be extended to integrate other types of sensitive channels.
>
> | Dataset  | Vanilla | KIVI  |  OTT  | ChanMix |
> | :------: | :-----: | :---: | :---: | :-----: |
> | AIME2024 |  43.33  | 26.67 | 16.67 |  26.67  |
> | AIME2025 |  23.33  | 16.67 |  20   |   20    |
>
> **Reference:**
> [1] Fu, Yu, et al. "Not All Heads Matter: A Head-Level KV Cache Compression Method with Integrated Retrieval and Reasoning." *The Thirteenth International Conference on Learning Representations*.
>
> > **Q3** The retrieval-head detection hyperparameters (n, t) are fixed “by experience.” However, stability across different values and architectures is not explored.
>
> Thank you for raising this concern about the stability of the hyperparameters (n,t) used in our retrieval-head detection method. We would like to clarify that our goal is to provide a general and robust approach for identifying retrieval heads, rather than to optimize hyperparameter values for each individual model.
>
> In practice, we adopt a single fixed configuration across all evaluated models including the Llama, Mistral, and Qwen families, as well as both MHA and GQA architectures. The experimental results consistently demonstrate the effectiveness and robustness of this unified setup.
>
> We also note that prior works [1] [2] [3] on retrieval-head detection similarly emphasizes practical and effective methods without exhaustive tuning for hyperparameter optimality. Exploring optimal configurations is indeed an interesting direction, and we consider it a promising future work.
>
> **Reference:**
>
> [1] Wu, Wenhao, et al. "Retrieval Head Mechanistically Explains Long-Context Factuality." *The Thirteenth International Conference on Learning Representations*.
>
> [2] Tang, Hanlin, et al. "RazorAttention: Efficient KV Cache Compression Through Retrieval Heads." *The Thirteenth International Conference on Learning Representations*.
>
> [3] Xiao, Guangxuan, et al. "DuoAttention: Efficient Long-Context LLM Inference with Retrieval and Streaming Heads." *The Thirteenth International Conference on Learning Representations*.
>
> > **Q4** Please clarify whether “%” refers to percentage points or relative percent.
>
> Thank you for pointing out the ambiguity. We have revised the wording to "5 absolute percentage point improvement" to avoid confusion in the updated PDF.

---

### Official Review · Reviewer_6Vs6 · 2025-10-31

**Soundness:** 3
**Presentation:** 3
**Contribution:** 3
**Rating:** 6
**Confidence:** 3

**Summary:**

This paper addresses the KV cache memory bottleneck for long-context inference. It identifies that the performance degradation of uniform low-bit quantization stems from asymmetric channel sensitivity. The authors propose ChanMix, a mixed-precision quantization framework that allocates bits based on a three-way channel classification: retrieval-sensitive channels (e.g., 4-bit), outlier channels (e.g., 3-bit), and robust/subnormal channels (e.g., 1-bit). The paper introduces an efficient one-shot method to identify these critical retrieval channels.

ChanMix is implemented with custom Triton kernels for efficient, 8-bit-aligned storage and dequantization. Experiments on Llama, Mistral, and Qwen show state-of-the-art results on RULER, NIAH, and InfiniteBench, significantly outperforming prior quantization methods.

**Strengths:**

- The paper's core strength is the novel insight that "retrieval-sensitive" channels are a distinct category from magnitude-based "outlier" channels, and both require higher precision . This three-way sensitivity (retrieval, outlier, subnormal) is well-motivated by analysis and validated by strong ablation studies .
- The proposed one-shot method for identifying retrieval heads is simple and efficient . The SOTA results (particularly the >5% gain on RULER) are significant.
- I like the custom Triton kernel implementation part, which fuses channel reordering with (de)quantization, makes the method highly practical and efficient

**Weaknesses:**

- The primary weakness is the heuristic nature of the bit allocation policy (1, 2, 3, 4 bits). The paper does not provide an analysis of how this policy was derived or its optimality.
- The generalizability of the one-time, offline channel profile is not thoroughly tested. It is unclear if a profile from Wikitext and a synthetic prompt  holds for all downstream tasks.
- The paper asymmetrically analyzes the K cache (channel-wise) but not the V cache (token-wise), lacking justification for this design choice.

**Questions:**

- How was the specific bit allocation (1, 2, 3, 4 bits) determined, and how sensitive is the model to changes in this policy?
- How robust is the offline channel profile? Does a profile generated on Wikitext transfer to specialized domains (e.g., code, math), or is reprofiling required for optimal performance?
- Why is the V cache not analyzed for channel-sensitivity and quantized channel-wise, similar to the K cache? Is the V cache less sensitive, or is this a design choice for simplicity?

I will raise my score if authors give a good rebuttal.

---

> ### Author Response · Authors · 2025-11-25
> **Response to Reviewer 6Vs6**
>
> We sincerely thank the reviewer for the insightful feedback and constructive suggestions. Below, we address the concerns in more detail.
>
> > **Q1** The primary weakness is the heuristic nature of the bit allocation policy. How was the specific bit allocation (1, 2, 3, 4 bits) determined, and how sensitive is the model to changes in this policy?
>
> Thank you for your critical question regarding the derivation and sensitivity of our bit allocation policy. We want to clarify that our strategy is not heuristic; it is a systematic design determined by engineering constraints and data-driven sensitivity analysis.
>
> The following results show that 4 bit quantization (ChanMix4) performs close to full precision, while 2 bit quantization (ChanMix2) still has noticeable degradation. Since ChanMix (and prior work like KIVI) specifically targets the 2 bit regime, our bit-reallocation strategy focuses on configurations at or below 4 bits.
>
> In our setting, {1,3} is used for subnormal and outlier channels, {2} is the base setting, and {4} is for retrieval channels. To fully utilize the memory storage, our allocation strategy must meet the 8 bit aligned storage (Refer to Figure 3 Right).
>
> - Subnormal & Outlier channels: As discussed in Section 4.1, outlier channels are more sensitive than subnormal ones, so we use the {1,3} configuration. We avoid {0,4} because it loses information, and combinations like {1,4} or {2,3} break the 8-bit alignment constraint.
> - Retrieval channels: As discussed in Section 4.1, retrieval channels require more bits, so we use 4 bits. We do not use 3 bits because this also fails to meet the alignment constraint.
>
> Overall, the results show that ChanMix2+O ({1,3}) improves accuracy without extra memory, ChanMix+R ({4}) also boosts performance, and the combined {1,2,3,4} configuration in ChanMix provides the best balance between storage efficiency and accuracy.
>
> |   Method   | KV Size |  Avg  |
> | :--------: | :-----: | :---: |
> |  ChanMix2  | 16.03%  | 82.20 |
> | ChanMix2+O | 16.03%  | 83.30 |
> | ChanMix2+R | 19.48%  | 83.35 |
> |  ChanMix   | 19.48%  | 83.87 |
> |  ChanMix4  | 28.53%  | 84.43 |
> |    FP16    |  100%   | 84.55 |
>
> \* Experiments are conducted on RULER-32K using Llama-3.1-8B-Instruct. ChanMix2 and ChanMix4 denote 2 bit and 4 bit quantization, respectively. +O indicates the 1,3 bit outlier compensation module, and +R denotes the 4 bit retrieval channel enhancement.
>
> > **Q2** How robust is the offline channel profile? Does a profile generated on Wikitext transfer to specialized domains (e.g., code, math), or is reprofiling required for optimal performance?
>
> We profiled the channel on WikiText with one-shot inference and then evaluated ChanMix on diverse downstream tasks across different domains MMLU (general knowledge), MBPP (code), and GSM8K (math) using two different models (Llama-3.1-8B-Instruct and Mistral-7B-Instruct-v0.3).
>
> ChanMix consistently maintains strong performance across all these tasks, which demonstrates the generalizability of our one-shot profiling method.
>
> |  Model  | Method  | MMLU  | MBPP | GSM8K |
> | :-----: | :-----: | :---: | :--: | :---: |
> | Mistral |  FP16   | 60.17 | 36.4 | 43.59 |
> | Mistral |  KIVI   | 59.82 | 29.4 | 39.19 |
> | Mistral | ChanMix | 60.15 | 33.4 | 42.46 |
> | Llama3  |  FP16   | 62.77 | 47.4 | 76.19 |
> | Llama3  |  KIVI   | 62.57 | 44.2 | 74.75 |
> | Llama3  | ChanMix | 62.54 | 47.2 | 73.54 |
>
> > **Q3** Why is the V cache not analyzed for channel-sensitivity and quantized channel-wise, similar to the K cache? Is the V cache less sensitive, or is this a design choice for simplicity?
>
> Precious works [1] [2] have shown that in key cache, some fixed channels exhibit very large magnitudes, whereas in value cache, there is no significant pattern for outliers. We also provide a visiualization of key and value cache on Appendix A.4 with updated PDF file, this phenomenon suggessts that we can apply channel-wise quantization for key cache, while it's not necessary for value cache.
>
> **Reference:**
>
> [1] Hooper, Coleman, et al. "Kvquant: Towards 10 million context length llm inference with kv cache quantization." *Advances in Neural Information Processing Systems* 37 (2024): 1270-1303.
>
> [2] Liu, Zirui, et al. "KIVI: A Tuning-Free Asymmetric 2bit Quantization for KV Cache." *International Conference on Machine Learning*. PMLR, 2024.

---

### Meta-Review · Area_Chair_uzma · 2025-12-27

**Summary:**

The paper proposes ChanMix, a mixed-precision KV cache quantization framework leveraging channel sensitivity. The review process was productive, with the authors effectively addressing concerns about reproducibility, experimental scope, and specific claims. Notably, this AC thinks that the negative review (Score 2) was primarily based on the initial lack of code and limited experimental settings rather than fundamental flaws. The authors resolved these issues by releasing code and conducting extensive ablation studies. Consequently, the discussion with the other reviewers turned highly positive.

**Reviewer Concerns:**

**Addressed**
 - Reproducibility (Reviewer oLua): The lack of code was a major blocking issue for the negative reviewer. The authors provided an anonymous GitHub link, resolving this concern.
 - Limited Experimental Settings (Reviewer oLua): The reviewer criticized the fixed group size and residual length. The authors added comprehensive ablation studies (9 combinations), proving the method's robustness.
 - Long-Output Performance (Reviewer oLua): Concerns about generation tasks were addressed by adding AIME benchmark results, showing superior performance.
 - Overstatements & Scalability (Reviewer HDr3): Claims regarding uniform quantization error were softened. Additional experiments on large models (Llama-70B, Qwen-32B) demonstrated scalability.

**Outstanding** None. This AC thinks the authors effectively addressed all major critiques.

**Reviewer Scores:**

- Reviewer HDr3: This reviewer actively engaged in the discussion and explicitly stated, "I am now inclined to recommend acceptance" following the revisions. Their score would definitively increase to an Accept.
- Reviewer oLua: This reviewer gave a Reject (2) based on specific, remediable issues like missing code and narrow settings. As their review was somewhat limited in scope and the authors fully provided the requested code and additional data, the score would likely see a significant increase.
- Reviewer 6Vs6: Likely maintained or increased their positive score, as the authors successfully demonstrated domain robustness (Math, Code).

---

### Decision · Program_Chairs · 2026-01-26

Accept (Poster)